# Effect of Different Composition on Voltage Attenuation of Li-Rich Cathode Material for Lithium-Ion Batteries

**DOI:** 10.3390/ma13010040

**Published:** 2019-12-20

**Authors:** Jun Liu, Qiming Liu, Huali Zhu, Feng Lin, Yan Ji, Bingjing Li, Junfei Duan, Lingjun Li, Zhaoyong Chen

**Affiliations:** 1College of Materials Science and Engineering, Changsha University of Science and Technology, Changsha 410114, Chinaliuqiming@stu.csust.edu.cn (Q.L.); junfei_duan@csust.edu.cn (J.D.); lingjun.li@csust.edu.cn (L.L.); 2College of Physics and Electronic Science, Changsha University of Science and Technology, Changsha 410114, China

**Keywords:** Li-rich layered oxide, cathode materials, voltage attenuation, lithium-ion batteries, solid-state complexation method

## Abstract

Li-rich layered oxide cathode materials have become one of the most promising cathode materials for high specific energy lithium-ion batteries owning to its high theoretical specific capacity, low cost, high operating voltage and environmental friendliness. Yet they suffer from severe capacity and voltage attenuation during prolong cycling, which blocks their commercial application. To clarify these causes, we synthesize Li_1.5_Mn_0.55_Ni_0.4_Co_0.05_O_2.5_ (Li_1.2_Mn_0.44_Ni_0.32_Co_0.04_O_2_) with high-nickel-content cathode material by a solid-sate complexation method, and it manifests a lot slower capacity and voltage attenuation during prolong cycling compared to Li_1.5_Mn_0.66_Ni_0.17_Co_0.17_O_2.5_ (Li_1.2_Mn_0.54_Ni_0.13_Co_0.13_O_2_) and Li_1.5_Mn_0.65_Ni_0.25_Co_0.1_O_2.5_ (Li_1.2_Mn_0.52_Ni_0.2_Co_0.08_O_2_) cathode materials. The capacity retention at 1 C after 100 cycles reaches to 87.5% and the voltage attenuation after 100 cycles is only 0.460 V. Combining X-ray diffraction (XRD), scanning electron microscope (SEM), and transmission electron microscopy (TEM), it indicates that increasing the nickel content not only stabilizes the structure but also alleviates the attenuation of capacity and voltage. Therefore, it provides a new idea for designing of Li-rich layered oxide cathode materials that suppress voltage and capacity attenuation.

## 1. Introduction

Advanced lithium-ion batteries (LIBs) technology have promoted the rapid development of mobile electronic devices owning to their low cost, long life, lack of any memory effect and environmental friendliness [1,2,3]. The development of plug in hybrid electric vehicles (PHEVs) and electric vehicles (EVs) puts higher demands on energy density and cruising range [4,5,6]. Compared to traditional cathode materials, such as LiCoO_2_ [7,8], spinel LiMn_2_O_4_ [9,10], polyanionic LiFePO_4_ [11,12], and layered cathode materials LiMO_2_ [13,14,15,16] (M = Ni, Co, Mn), Li-rich layered oxide cathode materials, represented by the chemical formula xLi_2_MnO_3_·(1−x)LiMO_2_ (0 < x < 1, M = Ni, Co, Mn) have received extensive attention from scientists all over the world because they exhibit a reversible capacity exceeding 250 mAh·g^−1^ between 2 V and 4.8 V, as well as their low cost and high energy density (>900 Wh·Kg^−1^) [17,18]. Unfortunately, these cathode materials put up with poor kinetics [19] and severe voltage attenuation [20,21,22] during prolong cycling, which directly affects their electrochemical performance, particularly the energy that the battery can output [23,24,25]. These disadvantages hinder the commercial development of high specific energy lithium-ion cells with cathodes prepared with the use of lithium-manganese-rich materials.

In order to explore effective ways to alleviate the voltage attenuation of Li-rich cathode materials, scientists have done a lot of research work to identify the origin [6,26,27,28]. The voltage attenuation can be caused by voltage fade and increase of resistance during cycling [29]. The cause of voltage attenuation is generally attributed to a continuous phase transition from layered phase to spinel during the repeated extraction/insertion processes [19,21,30], corresponding to the irreversible migration of the transition metal (TM) ions, during the course of which they move from the octahedral sites in the TM slab to the octahedral sites in the Li slab during the lithium ion extraction/insertion processes [31]. It is possible for Li-rich layered oxide cathode materials to prevent collapse of the layered structure by minimizing the tension between the neighboring oxygen layers in its deep delithiation state. In addition, another important reason for the voltage attenuation is increase of resistance during cycling [29]. Therefore, the voltage attenuation caused by resistance growth can be reflected in the average voltage as a function of cycling. Researchers believe that surface structure modification and elemental doping can effectively suppress voltage attenuation. For example, there are many reports of interface modifications (Al_2_O_3_ [32], LiFePO_4_ [33], LIPON [32], etc.) will reduce the side reaction between the surface of the positive electrode material and the electrolyte, and enhance the stability of the surface layered structure. elemental doping (Na^+^ [34], Ce^3+^/Ce^4+^, and Sn^4+^ [22], etc.) may significantly inhibit the Phase transition and stabilize the structure. Furthermore, Burrell et al. explored the effect of cycling temperature on the voltage fade of Li-rich layered oxide cathode materials [35]. Increasing the nickel content can effectively suppress the voltage attenuation in these cobalt-free or high-nickel cathode materials, which is the development trend of lithium-ion batteries in the future [23,25,36]. Dahn et al. reported Core-shell (CS) materials allow that the Mn-rich shell can protect the Ni-rich core from electrolyte attack, as well as the Ni-rich core maintains a high and stable average voltage [27,37]. Li et al. reported that reducing the Co content can significantly suppress voltage attenuation [38]. Vu et al. reported study the effect of composition on the voltage fade of Li-rich cathode materials with a combinatorial synthesis approach. Although the voltage fade can be reduced by controlling the composition of the system, there is no guarantee that the composition is the layered-layered structure [39]. Therefore, it is very important indication for Li-rich layered oxide cathode material to control the composition of TM ions in the LiMO_2_ layer in order to suppress the voltage attenuation.

To further investigate the effect of different compositions in the LiMO_2_ layer on the physicochemical properties of Li-rich layered oxide cathode materials, we synthesize Li_1.5_Mn_0.66_Ni_0.17_Co_0.17_O_2.5_ (Li_1.2_Mn_0.54_Ni_0.13_Co_0.13_O_2_) (LL-111), Li_1.5_Mn_0.65_Ni_0.25_Co_0.1_O_2.5_ (Li_1.2_Mn_0.52_Ni_0.2_Co_0.08_O_2_) (LL-523) and Li_1.5_Mn_0.55_Ni_0.4_Co_0.05_O_2.5_ (Li_1.2_Mn_0.44_Ni_0.32_Co_0.04_O_2_) (LL-811) cathode materials by means of solid-state complexation method, and the electrochemical properties have been investigated. During these three composition samples, LL-811 exhibits slower voltage decay and more excellent cycle stability during prolong cycling. The capacity retention rate of the LL-811 is 87.5% at 1 C rate after 100 cycles, and its voltage attenuation is quite low with 0.460 V. The distinctive advantage of this LL-811 Li-rich cathode material may derive from the high nickel content in the layered (R-3m) phase. High-nickel-content Li-rich layered oxide cathode material may cause more Ni^2+^ ions to occupy the Li^+^ ion sites in the lithium layer, it can result in a part of nickel to be doped at Li^+^ ion sites [40]. The cation doping, to some extent, can improve the structural stability by supporting the Li slabs and reducing tension of neighboring oxygen layers during the delithiation process [23]. In addition, nickel acts as a stabilizer to reduce the complete transformation of manganese by substitution. The preferential reduction of Ni^4+/2+^ maintains average oxidation state of Mn above 3+, as well as suppresses the Jahn-Teller effect caused by Mn^3+^ ions and effectively improves structural durability [41,42,43].

## 2. Materials and Methods

### 2.1. Sample Preparation 

The Li_1.5_Mn_0.66_Ni_0.17_Co_0.17_O_2.5_ (Li_1.2_Mn_0.54_Ni_0.13_Co_0.13_O_2_), Li_1.5_Mn_0.65_Ni_0.25_Co_0.1_O_2.5_ (Li_1.2_Mn_0.52_Ni_0.2_Co_0.08_O_2_), Li_1.5_Mn_0.55_Ni_0.4_Co_0.05_O_2.5_ (Li_1.2_Mn_0.44_Ni_0.32_Co_0.04_O_2_) cathode materials (marked as LL-111, LL-523, LL-811, respectively.) were synthesized by a solid-sate complexation method using citric acid monohydrate as complexing agent with analytical grade chemicals LiAc·2H_2_O (Excess 5%,AR, 99%), Ni(Ac)_2_·4H_2_O (AR, 98%), Co(Ac)_2_·4H_2_O (AR, 99.5%), Mn(Ac)_2_·4H_2_O (AR, 99%), citric acid monohydrate (AR, 99.5%). The molar ratio between transition metal ion and citric acid monohydrate was 1:1. Using a certain amount of absolute ethanol as a solvent, a stoichiometric amount of above reagents were mixed thoroughly and ball milled at the speed of 200 rpm continuously for 4 h. After ball milling, it was dried in an oven at 80 °C for 24 h in order to get a uniform mixed precursor. The precursor was ball milled for 30 min and precalcined at 450 °C for 6 h in air to eliminate the organic substances then was calcined at 900 °C, 900 °C, 800 °C for 12 h in air at the rate of 5 °C·min^−1^, respectively. And finally, these required samples were obtained.

### 2.2. Materials Characterizations

The crystallographic structure LL-111, LL-523 and LL-811 cathode materials were carried out by X-ray diffraction (XRD, Bruker D8, Karlsruhe, Germany) with Cu Kα radiation (λ = 1.54056 Å) in the range of 10–90° with the speed of 5° min^−1^. The microscopic morphology was investigated with scanning electron microscopy (SEM, TESCAN MIRA3 LMU, Brno, Czech Republic) and transmission electron microscopy (TEM, TECNAI G2 F20, Hillsboro, America).

Electrochemical performance of the samples was characterized using galvanostatic charge-discharge tests with two-electrode coin cells (type CR-2025, Shenzhen, China). All the charge-discharge processes of our CR-2025 cells except for the first cycle was measured under a voltage window of 2–4.6 V. The synthesized sample, acetylene black (AR, Hersbit Chemical Co., Ltd., Shanghai, China) and polyvinylidene difluoride (PVDF, FR905, San ai fu New Material Technology Co., Ltd., Shanghai, China) with a weight ratio of 8:1:1 to make a slurry in the N-methyl pyrrolidone (NMP) solvent. The slurry was uniformly coated onto aluminum foil as current collector and then dried at 120 °C for 6 h under vacuum oven. Cells were assembled in an Argon-filled glove box with H_2_O and O_2_ contents below 0.01 ppm, using the metallic lithium foil as an anode. The electrolyte was 1 M LiPF6 dissolved in ethyl carbonate (EC) and dimethyl carbonate (DMC) (1:1 in volume) and the separator was Celgard-2500 membrane.

The Galvanostatic charge-discharge measurements were carried out using NEWARE CT-4008 battery testing system (Shenzhen, China) within the voltage range of 2.0–4.8 V at 25 °C. Cyclic voltammetry (CV) and AC impedance (EIS, 1 MHz–0.1 MHz) using a CHI660E electrochemical workstation (Shanghai, China).

## 3. Results and Discussion

### 3.1. Crystal Structure and Microstructure

Using citric acid monohydrate as complexing agent, the cathode materials LL-111, LL-523, and LL-811 were synthesized by a solid-state complexation method. The phase and crystal structure of the above cathode materials were respectively analyzed by XRD. The XRD patterns of the sample LL-111, LL-523, LL-811 are shown in Figure 1. Using the JADE 6.5 analysis software to analyze the XRD data, the diffraction peaks of three samples can be indexed to a typical α-NaFeO_2_ structure (R-3m) and the enlargements of the small region from 20 to 24° are attributed to Li_2_MnO_3_ phase (C2/m) [44,45]. Complete splitting of the two pair diffraction peaks (006)/(012) and (018)/(110) indicates the integrity of the layered structure for all samples. It is noted that, the intensity of Li_2_MnO_3_ characteristic peak gradually decreases with the metallic nickel content increasing.

The surface morphologies of LL-111, LL-523, LL-811 samples and corresponding element compositions are shown in Figure 2. As seen in Figure 2a–c, three samples show irregular polyhedral morphology with a size of approximately 200–500 nm, which is consistent with previous literatures [6,46,47]. In Figure 2d–f, the primary particles of three samples show different degrees of agglomeration. As the nickel content increases, particle agglomeration becomes more and more seriously. Besides, the interface between the particles is blurred and even disappears. As seen in Figure 2g–i, the actual ratios of LL-111 and LL-523, LL-811 cathode materials are basically in accordance with the designed values. However, LL-811 (Figure 2i) shows a Mn-rich trend on the surface, which may be mainly due to the concentration gradient of Ni/Mn with the nickel content increasing.

To explore the precise structural properties of Li-rich cathode materials with high nickel content, TEM and HRTEM images are shown in Figure 3. As seen in Figure 3a,d,g the edge of LL-811 cathode material is evenly distributed. Besides, it has relatively straight and continuous lattice fringes, and the interplanar spacing of (003) crystal plane is 4.72 Å (Figure 3b), which shows the characteristics of good layered structure material (FFT results in insets of Figure 3c). The lattice fringes of Figure 3e show a distinct two-phase composite (FFT results in insets of Figure 3f). As clearly seen in Figure 3g, the internal phase distribution of the bulk is not very uniform, and Figure 3h further verifies this phenomenon. Figure 3h and FFT results show three kinds of different plane spaces and crystal plane orientations, which represent the (003) crystal plane of the LL-811 structure (4.72 Å), the (006) crystal plane of the LL-811 structure (2.36 Å), two-phase (Li_2_MnO_3_ and LiMO_2_) composite, respectively.

In order to further verify the elemental uniformity of the LL-811 sample, the EDX linear scanning of the single particle was carried out in Figure 4a,b. A very important message can be obtained from Figure 4b, in which the content of Mn element gradually increases from internal to external and Mn-rich phase appears on the surface. However, the change trend of nickel element is opposite, which may be mainly due to the segregation of Mn and Ni elements. The segregation may be induced by citric acid and be also due to the bond formation of the layered material itself [48,49]. Nickel has high catalytic activity and is easy to react with electrolyte, while Mn-rich on the surface can effectively inhibit the reaction. The Mn-rich surface phenomenon may explain the excellent prolong cycling stability of LL-811 cathode materials [25,27].

### 3.2. Electrochemical Charge/Discharge Behavior 

In order to evaluate the electrochemical property of the samples LL-111, LL-523, and LL-811, the first charge-discharge curves, rate performances from 0.1 C to 5 C and cycling performances at 1 C (1 C = 200 mAh·g^−1^) between 2 V to 4.8 V are shown in Figure 5. As shown in Figure 5a, there are two distinct voltage plateaus during the first charge process: (1) a smooth voltage plateau below 4.5 V and (2) a long voltage plateau about 4.5 V [50]. The first discharge capacities of LL-111, LL-523, LL-811 are 284.6 mAh·g^−1^, 263.0 mAh·g^−1^, 207.4 mAh·g^−1^, respectively.

LL-111 samples at 1 C delivers 206.4 mAh·g^−1^ and about 98.5% of the capacity is maintained even after 100 cycles. The first discharge capacity of LL-523 at 1 C is 194.1 mAh·g^−1^ and about 80.0% of the capacity is maintained after 100 cycles. In contrast to LL-111 and LL-523, the capacity retention of the LL-811 is 87.5% after 100 cycles, although the first discharge capacity of LL-811 at 1 C only is 154.6 mAh·g^−1^. By comparison, the cycle stability of the high-nickel-content LL-811 cathode material is more excellent. It can be attributed to that nickel ions easily migrate out of the TM layer to support the structure instead of being trapped in the middle tetrahedral layer [23].

Figure 5c shows the rate performance of these three samples at different current densities of 0.1 C, 0.2 C, 0.5 C, 1 C, 3 C, 5 C at 25 °C between 2 V and 4.8 V. As seen in Figure 5c, the discharge capacities of LL-111, LL-523, and LL-811 are 130.8 mAh·g^−1^, 118.9 mAh·g^−1^, and 100.5 mAh·g^−1^, respectively at 5 C. The low rate capacity is ascribed to the sluggish kinetics of high nickel Li-rich layered oxide cathode materials discussed in the present paper. The method of calculating the lithium ion diffusion coefficient by EIS measurement is the same as the previous paper [51]. After EIS measurement, Li^+^ diffusion coefficient of LL-111, LL-523, and LL-811 three samples are 2.77 × 10^−14^ cm^2^ S^−1^, 3.70 × 10^−14^ cm^2^ S^−1^, and 1.32 × 10^−14^ cm^2^ S^−1^, respectively. Moreover, as seen in Figure 5g, the charge transfer impedance (Rct) is 212 Ω, 352 Ω, and 606.7 Ω for the LL-111, LL-523, and LL-811 samples. The Rct of high nickel LL-811 cathode material is very large compared to the other two samples, which can be attributed to the low electronic conductivity.

The most striking performance feature of LL-811 cathode materials is its low voltage attenuation after prolong cycling. Figure 5e shows the relationship between the average voltage and cycle number. The voltage attenuation is approximately 0.460 V after 100 cycles at 1 C, while for LL-111 and LL-523, the voltage attenuation is reached up to 0.665 V and 0.600 V, respectively.

To further illustrate the voltage attenuation phenomenon of Li-rich cathode materials upon cycling, discharge curves with different cycles between 2 V to 4.8 V at 1 C and cyclic voltammetry (CV) curves of LL-111, LL-523, and LL-811 are shown in Figure 6. As seen in Figure 6a–c, voltage attenuates rapidly at 1 C for LL-111, and for LL-523 it is slower than LL-111, whereas for LL-811 it is the lowest. Figure 6d–f clearly characterize that the three samples display distinct oxidation peaks appeared at 4.0 V during the first cycle, corresponding to the oxidation reactions of Ni^2+/3+/4+^ and Co^3+/4+^; and the oxidation peak at 4.6 V corresponds to the activation process of Li_2_MnO_3_ [52]. The reduction peaks at 3.3 V, 3.6 V, 4.1 V reflect Mn^4+/3+^, Ni^4+/3+^, Co^4+/3+^ [53], respectively. By comparison, as seen clearly in Figure 6d–f, the voltage decay is minimal for LL-811 after 3 cycles, which is consistent with the phenomenon observed with the voltage attenuation curves. LL-111 and LL-523 Li-rich materials suffers from voltage attenuation after prolong cycling, especially LL-111, for which its capacity mainly came from the low voltage region [23]. Therefore, the specific energy (specific energy = specific capacity × average voltage) output of the battery further lowered upon cycling owing to the disappointing cycle stability and severe voltage attenuation for LL-111 and LL-523 cathode materials. This is due to the phase transformation from layered to spinel-like or rock-salt phases during repeated charge and discharge cycles. As seen in Figure 5d, the absolute value of specific energy of LL-111, LL-523, and LL-811 cathode materials at the 100th cycle is 474.82, 465.88, 435.68 Wh·Kg^−1^, respectively. Comparing with LL-111 and LL-523 (64.6% and 67.0% energy retention after 100 cycles, respectively), the specific energy retention was 76.7% for LL-811 after 100 cycles. The excellent cell properties of LL-811 indicate that increasing the nickel content could significantly inhibit the intrinsic voltage attenuation of Li-rich materials. These consequences demonstrate that high-nickel-content Li-rich cathode materials, such as LL-811, exhibit outstanding structural durability during prolong cycling, which will promote the commercialization of Li-rich cathode materials.

To further assess the effect of nickel content on the structural durability of Li-rich cathode materials, the XRD patterns of LL-111, LL-523, LL-811 before and after 100 cycles are shown in Figure 7. The intensity ratios of both the (003) and (104) peaks of LL-111, LL-523, and LL-811 are shown in Table 1. Compared to LL-111 and LL-523, the intensity ratio of the (003) and (104) peaks of the LL-811 cathode material is still greater than 1.2 after 100 cycles [54,55], which indicates that the high nickel cathode materials can maintain their structural durability and inhibit the phase transformation to a certain degree during prolong cycling.

In this paper, Li-rich layered oxide cathode materials with different compositions are designed and prepared. From the above discussion results, the sensitivity of the voltage attenuation phenomenon to the composition is relatively large.

## 4. Conclusions

In this paper, LL-111, LL-523, and LL-811 cathode materials were successfully synthesized by a solid-sate complexation method using citric acid monohydrate as complexing agent. Compared to the LL-111 and LL-523, the high-nickel-content LL-811 cathode material shows an excellent cycling stability (capacity retention of 87.5% at 1 C rate after 100 cycles and energy retention of 76.7% at 1 C rate after 100 cycles) and suppresses voltage attenuation (only 0.460 V after 100 cycles) during prolong cycling. CV curves also show that the high-nickel-content LL-811 cathode material exhibits less polarization during cycling. What is more, cycled XRD results demonstrate that increasing the nickel content can effectively maintain structural stability. The results show that the sensitivity of the voltage attenuation to the composition is relatively large. This proves that it is very important for Li-rich layered oxide cathode materials to control the composition of TM ions in the LiMO_2_ layer. This further demonstrates that the nickel content plays a very important role in stabilizing the structure and suppressing the voltage attenuation. This paper provides a reference for the composition design of Li-rich layered oxide materials with the high nickel content.

## Figures and Tables

**Figure 1 materials-13-00040-f001:**
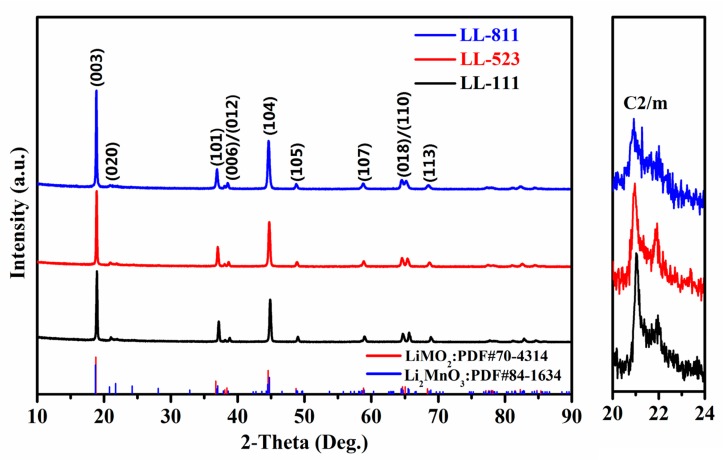
XRD patterns of LL-111, LL-523, LL-811, which adopts a α-NaFeO_2_ structure. The XRD patterns on the right show the enlargement of Li_2_MnO_3_ characteristic peak over a small 2θ region.

**Figure 2 materials-13-00040-f002:**
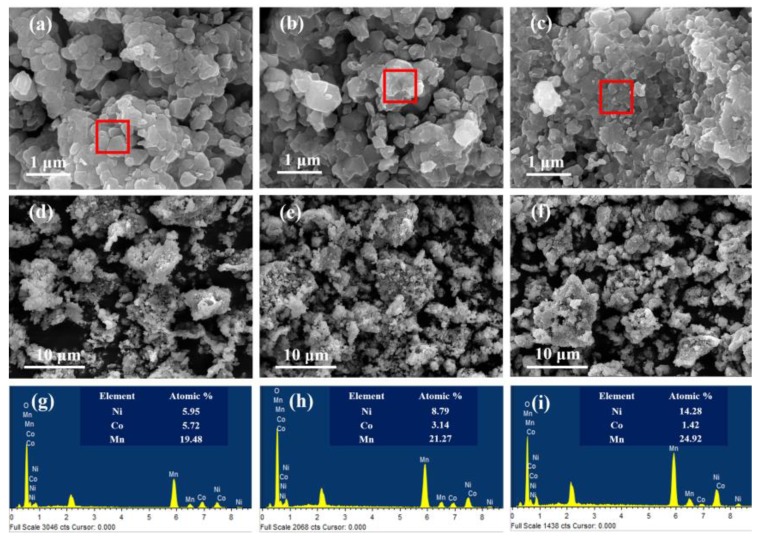
SEM images of LL-111 (**a**,**d**), LL-523 (**b**,**e**), LL-811 (**c**,**f**); Elemental mapping images (**g**–**i**) of LL-111, LL-523, LL-811, respectively.

**Figure 3 materials-13-00040-f003:**
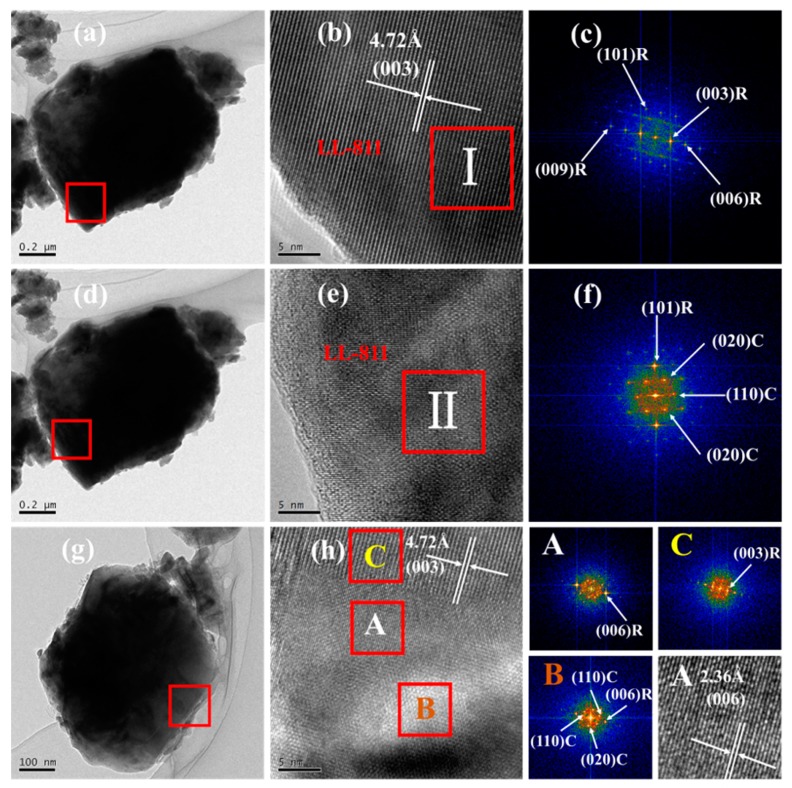
TEM images: (**a**,**d**,**g**), HRTEM images: (**b**,**e**,**h**) of selected regions of LL-811 and corresponding to the Fast Fourier transform (FFT) images of I and II regions: (**c**,**f**).

**Figure 4 materials-13-00040-f004:**
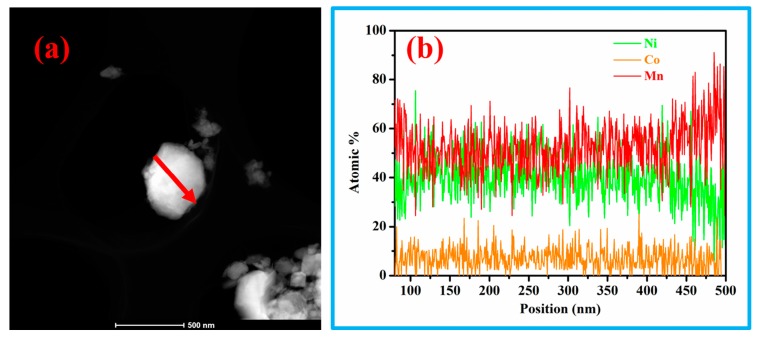
(**a**) TEM image of LL-811 single particle; (**b**) corresponding to the Energy dispersive X-ray linear scanning.

**Figure 5 materials-13-00040-f005:**
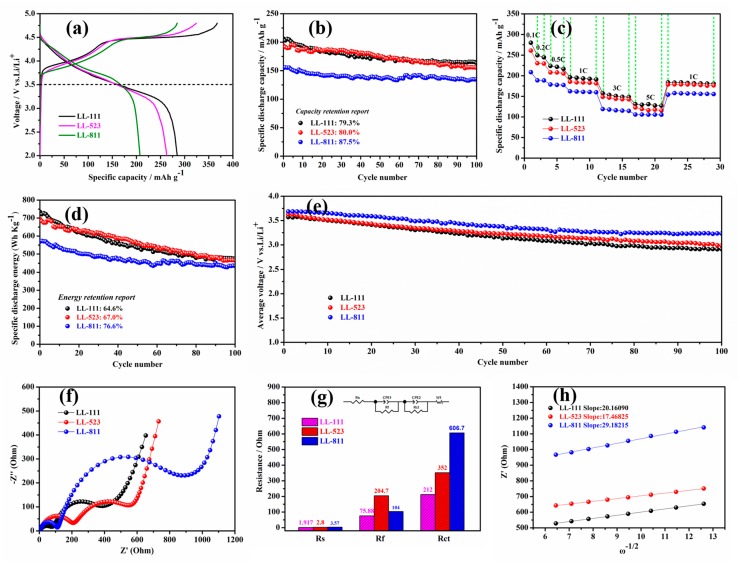
Electrochemical performances: (**a**) First charge- discharge curves at 0.1 C; (**b**) The cycle performances at 1 C; (**c**) The rate performances from 0.1 C to 5 C; (**d**) Specific discharge energy cures at 1 C; (**e**) Average voltage attenuation curves after 100 cycles at 1 C; Nyquist plots: (**f**) Fresh cells; (**g**) Fitted impedance data of LL-111, LL-523, and LL-811 cells between 2 V to 4.8 V; (**h**) The typical plots of Z’ vs. ω^−1/2^ at low frequency.

**Figure 6 materials-13-00040-f006:**
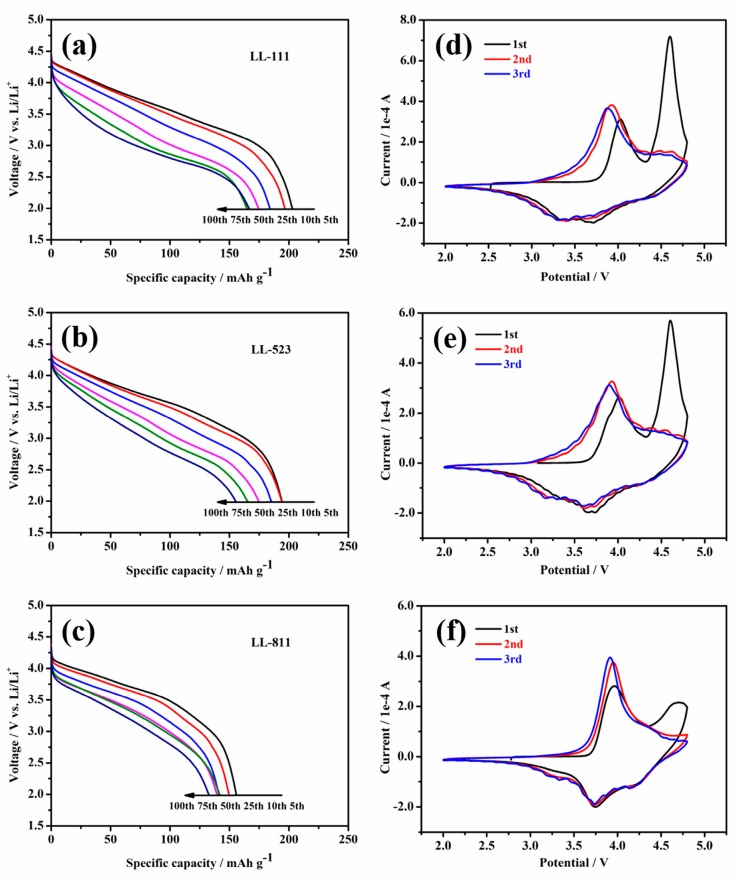
Discharge curves with different cycles between 2 V to 4.8 V at 1 C and Cyclic Voltammetry (CV) curves of LL-111 (**a**,**d**), LL-523 (**b**,**e**), and LL-811 (**c**,**f**).

**Figure 7 materials-13-00040-f007:**
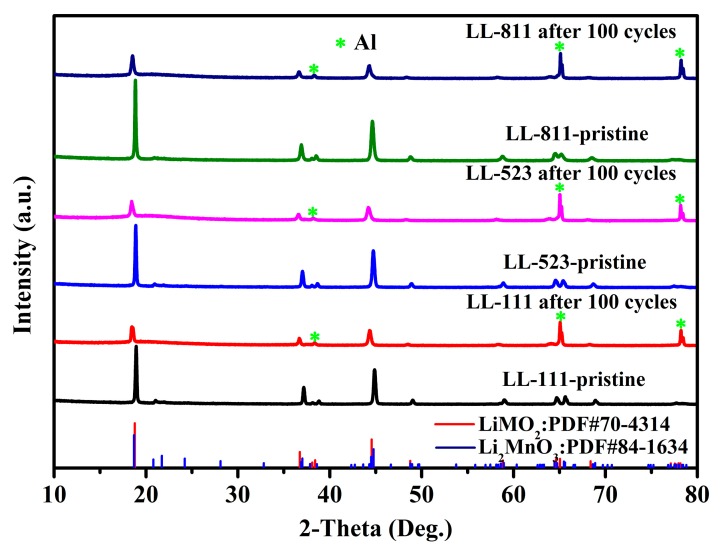
XRD patterns of LL-111, LL-523, LL-811 before and after 100 cycles.

**Table 1 materials-13-00040-t001:** The intensity of (003) and (104) and the ratios I_(003)_/I_(104)_ of LL-111, LL-523, and LL-811 initial samples and cycled electrodes by calculating from the XRD data.

Samples	I_(003)_	I_(104)_	Initial I_(003)_/I_(104)_	I_(003)_	I_(104)_	Cycled I_(003)_/I_(104)_
LL-111	6553	3941	1.6628	2215	1803	1.2285
LL-523	7017	4189	1.6751	2286	1611	1.4190
LL-811	9116	4531	2.0120	2670	1563	1.7083

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
