# Peer review of "Effect of Different Composition on Voltage Attenuation of Li-Rich Cathode Material for Lithium-Ion Batteries"

_materials, 2019, doi:10.3390/ma13010040_

Round 1

Reviewer 1 Report

Dear authors,

The reviewed manuscript has interesting results, but after reading carefully, I have several questions and suggestions.

Page 1, Line 13. (Global) a reviewer suggests using the term “high specific energy” instead of “high-energy-density” since energy density corresponds to full energy divided per volume of the cell. The lithium-rich materials have high specific capacity per gram. Because of relatively small density (therefore the density of electrodes active layer), the high-energy-density of the cell is unlikely to be achieved.

Page 1, Line 31. HEV cells need to possess high power density on specific power per kilogram, not energy density. At the worst, a reviewer suggests to change “HEVs” by “PHEVs” or eliminate a part of the sentence related to HEVs.

Page 1, Line 37. Note: From the point of view of battery (or device) developer, the range 4.8-2.0 is very broad.

Page 1, Line 41. Please replace “of high-energy-density lithium-ion batteries” by “high specific energy lithium-ion cells with cathodes prepared with the use of lithium-manganese-rich materials”. Or by part of the sentence with the same sense.

Page 1, Line 41 – Page 2, Line 57. Note: The voltage attenuation can be caused by “Voltage fade” and increase of resistance during cycling to distinguish the contribution of each process, please look at the following paper and presentation: 10.1149/2.034311jes, https://www.energy.gov/sites/prod/files/2014/03/f13/es188_abraham_2013_p.pdf

Page 2, Lines 42-57 – Could you please describe some methods used to suppress voltage attenuation specifically in lithium-rich materials. Besides searching in scientific databases, a reviewer suggests looking for papers mentioned in related project reports financed by DOE https://www.energy.gov/eere/vehicles/annual-merit-review-presentations (Project Leader Anthony Burrell and others from voltage fade team, 2012-2014).

According to a literature review, there was an article published several years ago related to your research. Anh Vu et al. studied the effect of lithium-manganese-rich cathode material composition on the voltage fade (10.1016/j.jpowsour.2015.06.100 + https://www.energy.gov/sites/prod/files/2014/07/f17/es190_johnson_2014_o.pdf Page 11). Could you please reveal the novelty of your research in comparison with research results presented in publication 10.1016/j.jpowsour.2015.06.100.

Page 2, Line 84. Please change “the organic substances and then was cancined” by “the organic substances and then was calcined”.

Page 2, Line 93, Page 3, Line 94. Could you please provide the name of the manufacturer as well as the mark for acetylene black and PVDF used for electrodes preparation.

Page 3, Line 121-123. Have particles formed the agglomerates? If yes, what was the size of agglomerates observed during SEM investigations?

Page 3, Line 125. A reviewer suggests to disclose the “designed values” and save time for the Reader for performing calculations with formulas. If the reviewer is not mistaken, the chemical formulas of lithium-rich materials can be written in two other ways: LL 811 – Li1.5Mn0.55Ni0.4Co0.05O2.5 (Li1.2Mn0.44Ni0.32Co0.04O2).

Page 4, Lines 129-130. Fig.2 (d,e,f) - Hard to read. Must be improved.

Page 4, Line 140. Li2MnO3 and NCM both have layered structures, and both have (003) and (006) reflexes in diffractogram. Could you please clarify why based on TEM results one can conclude the presence of two phases (Symmetry?)

Page 5, Lines 154-155. Fig.4b - Hard to read. Must be improved.

Li content in cathodes is not discussed. It would be good to use ICP-AES method.

Page 5, Lines 161-164. The sentence “The ionic radius… nickel content increases” this sentence is not related to the previous text in a paragraph. Furthermore, the cation mixing determined on the base of reflexes ((003), (104)) intensity seems to be absent (Table 1).

Page 6, Lines 175-176. “The low rate capacity is ascribed to the sluggish kinetics of high nickel cathode materials” From this sentence one can conclude that all high nickel cathode materials have bad kinetics. But high nickel cathode materials used in high power cells (Saft VL5U, Toyota Prius 4 etc.) thus presented assumption seemed to be odd. A reviewer suggests adding to the sentence “discussed in the present paper”.

Page 6, Lines 176-177, Page 102-103. Could you please describe the calculation procedure of the apparent diffuse coefficient.

Page 7, Lines 205-207. Could you please provide the absolute values of specific energy at the 100th cycle. The relative values suggest that the LL 811 performs better, but it might have smaller energy.

Page 7, description of Fig. 6. According to Fig 5a (lower charge/discharge capacity, smaller activation plateau during charging) and Fig. 6d, 6e, 6f (the intensity of anode peak at 4.6V, as well as the intensity of cathode peak at 3.2-3.4V is smaller) with an increase of Ni-content in NCM phase the content of Li2MnO3 phase in LL materials decreases, thus the drop of voltage corresponding to voltage fade process decreases. Does a part of Mn ions go to the NCM phase?

Page 8, Lines 222-225. Fig. 7 and Table 1. It would be nice to compare cycled electrodes with initial, not with powders.

In conclusions, the sentence "This paper promotes effect of different components on voltage attenuation of Li-rich cathode materials, especially the nickel content, which is of great reference and guiding significance for the practical application of Li-rich layered oxide cathode materials." is too common. The authors must provide more significant discussion and data.

The novelty is not clear. Please, please do more strong accent on it.

Reviewer 2 Report

This work presents tuning the composition of the Li-rich layered oxide cathode materials that suppress voltage and capacity attenuation of Li-ion batteries. Li-rich layered oxide cathode materials become one of the most promising cathode materials for LIB. This manuscript has been well documented from material synthesis to its electrochemical behavior with adequate physical characterization.   However, in its present form is not publishable and need a minor revision before getting published in Materials.

All the abbreviates should be explained in their first appearance, such as LL-111 and LL-523 in the abstract. The authors can consider using LL-333 or closely reasonable labels to index the 1/3 ratio. Respective JCPDS Standard reference of the XRD patterns should be provided. There is a conflict between the interpretation of SEM and TEM. Where in SEM said that nickel rich compound is agglomerated and LL-111 shows a smooth surface. But in TEM authors had claimed that the LL-811 has a smooth surface. In page 4, line 146, discussed” segregation of Mn and Ni elements”. Whether this segregation is induced by the citric acid or due to the bond formation of the layered material itself. Authors can consider using these suitable references (Electrochimica Acta 60 (2012)170-176; Energies 2019, 12, 1329) to support this discussion. “The ionic radius of Ni2+ (0.70Å) is 162 similar with Li+ (0.76Å), so it could result in some nickel ions occupying lithium sites as the nickel content increases” If so then the change in valance of these ions should display structural defects, any such effect is observed in XRD.

Round 2

Reviewer 1 Report

Dear authors,

Thank you for revising the manuscript. I satisfied with the answers and recommend accepting the manuscript for publication after minor corrections:

Point 2: please keep two forms of designation form of LL111, LL523, LL811 – previous one with two formulas used before revision, as well added in reviewer response.

Point 3: Please add “All the charge-discharge processes of our CR-2025 cells except for the first cycle was measured under a voltage window of 2-4.6V.” to the relevant paragraph of 2.2 section.
